# Dynamic Modeling of Weld Bead Geometry Features in Thick Plate GMAW Based on Machine Vision and Learning

**DOI:** 10.3390/s20247104

**Published:** 2020-12-11

**Authors:** Yinshui He, Daize Li, Zengxi Pan, Guohong Ma, Lesheng Yu, Haitao Yuan, Jian Le

**Affiliations:** 1School of Resources Environmental & Chemical Engineering, Nanchang University, Nanchang 330031, China; yshhe@ncu.edu.cn (Y.H.); 5801117023@email.ncu.edu.cn (D.L.); 2School of Mechanical, Materials, Mechatronic and Biomedical Engineering University of Wollongong, Wollongong, NSW 2500, Australia; zengxi@uow.edu.au; 3Key Laboratory of Lightweight and High Strength Structural Materials of Jiangxi Province, Nanchang 330031, China; mgh@ncu.edu.cn (G.M.); 400906318037@email.ncu.edu.cn (L.Y.); 410926719033@email.ncu.edu.cn (H.Y.); 4Information Engineering School of Nanchang University, Nanchang 330031, China

**Keywords:** weld bead geometry features, visual all-position measurement, thick plate gas metal arc welding, machine vision and learning, fault detection and diagnosis

## Abstract

Weld bead geometry features (WBGFs) such as the bead width, height, area, and center of gravity are the common factors for weighing welding quality control. The effective modeling of these WBGFs contributes to implementing timely decision making of welding process parameters to improve welding quality and enhance automatic levels. In this work, a dynamic modeling method of WBGFs is presented based on machine vision and learning in multipass gas metal arc welding (GMAW) with typical joints. A laser vision sensing system is used to detect weld seam profiles (WSPs) during the GMAW process. A novel WSP extraction method is proposed using scale-invariant feature transform and machine learning. The feature points of the extracted WSP, namely the boundary points of the weld beads, are identified with slope mutation detection and number supervision. In order to stabilize the modeling process, a fault detection and diagnosis method is implemented with cubic exponential smoothing, and the diagnostic accuracy is within 1.50 pixels. A linear interpolation method is presented to implement sub pixel discrimination of the weld bead before modeling WBGFs. With the effective feature points and the extracted WSP, a scheme of modeling the area, center of gravity, and all-position width and height of the weld bead is presented. Experimental results show that the proposed method in this work adapts to the variable features of the weld beads in thick plate GMAW with T-joints and butt/lap joints. This work can provide more evidence to control the weld formation in a thick plate GMAW in real time.

## 1. Introduction

The gas metal arc welding (GMAW) process has become a popular method for joining the thick plates of steel structures and been used in the shipbuilding industry, wire arc additive manufacturing [1], pipe manufacturing [2,3], etc. Automatic and intelligent welding technologies have been widely applied to the thick plate GMAW process, such as automatic seam tracking [4] and weld formation monitoring, to enhance welding quality and efficiency [5]. In thick plate GMAW, the online monitoring of weld formation for each pass is necessary for weldment quality control, and welding process parameters are deemed the main factors that exceedingly influence weld formation [6,7,8]. Weld bead geometry features (WBGFs) can reflect the welding process parameters and are effective evidence to suggest how to optimize the latter. This is because the relationship between WBGFs and weld process parameters can be built with various models [9,10,11]. Thus, the real-time modeling of WBGFs is the really demanded technology for the effective control of weld formation especially and accurate metal deposition [12,13] during the multipass welding process. 

The modeling process of WBGFs is implemented in an offline and online fashion. In some studies, the prediction or optimization of WBGFs is the main objective using regression models between WBGFs and welding process parameters. In this case, offline modeling methods are presented via the metallurgical microscope [14], the caliper [15], the optical microscope [16,17], and the machinist’s microscope [18] in lap-joint GMAW. These studies are dependent on the visual ability of the evaluator, where the measurement process is conducted with human intervention [19]. In addition, ultrasonic sensing [20] and infrared sensors are also used in the modeling process to control the depth of penetration and the bead width [21].

The vision sensing technology is one of the main online measurement methods of WBGFs and the most common method for welding position detection [22,23]. This kind of technology usually uses active or passive sensors to detect targets, and image processing methods are responsible for the extraction of WBGFs. During the measurement process with passive sensors, multiple cameras are used to respectively detect the bead width and height [24]. A single high-speed camera is used to acquire the related WBGFs combining with the image calibration technology [25]. For an active vision sensing measurement process, researchers generally use the laser to illuminate the weld bead to create a deformation that is proportional to the bead width and height. Hence, WBGFs are modeled by detecting the bead edges such as Canny edge detection [26,27].

With the development of machine vision and learning, laser vision sensing has been used in the welding industry because it can detect deep grooves with thick steel plates. This ability can measure more WBGFs. In these studies, the sensor emits a laser light with the specific wavelength onto the groove, and a charge coupled device (CCD) camera captures the image. If the sensor works statically, the two-dimensional measurement of WBGFs can be realized [28]. A three-dimensional WBGF measurement is implemented by moving the sensor along the welding direction [29,30,31] using image processing technologies.

It is noteworthy that the current studies mainly concentrate on modeling WBGFs for a single weld bead; most of them are implemented for lap joints, and WBGFs usually are the bead width and height. More WBGFs such as the area and the center of gravity are also useful evidence for weld formation control. In addition, the real-time modeling process of these WBGFs in thick plate GMAW is still a challenge, which demands higher adaptability of information acquisition, stable determination of the weld bead, and variable WBGFs. Thus, more effective weld seam profile (WSP) extraction and feature point identification methods, stable seam tracking technologies, and the effective modeling scheme of the WBGFs should be investigated. In addition, the WBGFs at different formation positions provide more direct evidence to optimize the welding process parameters (e.g., the angle of the welding torch) than only their maxima. 

WSP extraction is the prerequisite during the active-sensing-based WBGF modeling process. Various WSP extraction methods have been reported in the literature [32,33]. However, the extraction process of WSPs with larger sizes faces a bigger challenge during the multipass welding process because this process is easily disturbed. There is no doubt that an appropriate fault detection and diagnosis process [33] contributes on success a higher probability of visual feature acquisition results. The cubic exponential smoothing algorithm can predict future states of the research object using the past sampling for time series variables. It has been applied to different industries [34,35] for optimization or prediction purposes.

In this work, a laser vision sensor is used to detect the deep grooves with T-joints and butt joints during the multipass GMAW process. A more effective WSP extraction method is presented based on scale-invariant feature transform (SIFT) and machine learning algorithms to strengthen the robustness comparing with the previous ones [36,37]. This extraction method adapts to variable WSPs. The feature points of the extracted WSP are effectively identified through slope mutation detection and supervising the number. In order to stabilize the seam tracking and the subsequent modeling process of WBGFs, a fault detection and diagnosis process is applied to the feature point identification process via the cubic exponential smoothing method. A scheme of modeling WBGFs is proposed using the diagnosed feature points and the extracted WSP. With the proposed method in this work, the variable weld bead of the area, center of gravity, and all-position weld bead width (APWBW) and height (APWBH) are modeled in real time for different passes. Various experimental results show the effectiveness of the proposed method. The proposed method here contributes to implementing timely decision making of welding process parameters to improve the welding quality and enhance automatic levels.

This work consists of WSP extraction, feature point identification of the WSP, fault detection and diagnosis, modeling WBGFs, experimental results, discussion, and conclusions.

## 2. WSP Extraction with SIFT and Machine Learning

In order to monitor the WBGFs, especially the center of the gravity of the weld bead, simultaneous imaging of the welding torch and the WSP is investigated through a special combination of dimmer glass and the filter in this work. Experimental results show that when the central wavelength of the filter is about 660 nm, the half bandwidth is 20 nm, and the transmittance of the dimmer glass is about 0.02%, the welding torch can be marked with the welding wire. Figure 1a shows the typical work state of the sensor, and Figure 1b gives the imaging effect of the laser stripe and the wire. The direction detection of the welding torch contributes to weld formation control. Here, only the WSP extraction process is given as follows, and WSP extraction with T-joints is used as an example.

WSPs changes with the weld beads during the multipass GMAW process. However, the image (reference image) that is captured before arc starting has a similar laser stripe (the shape, the position, and the intensity) with the one captured after the welding process begins (raw image). Thus, the WSP extraction method is proposed based on SIFT and machine learning as shown in Figure 2, in which the SIFT algorithm can effectively match the characteristics between the two images.

This method includes two image processing paths. The first one is using the Gabor filter and local thresholds to highlight the WSP from the background and simplify the background data, respectively. The second one is locating the WSP from the interference data based on SIFT and the nearest neighbor clustering algorithm.

The first image processing path is as follows. The region of interest (ROI) is normally used to reduce the influence of the interference data on WSP extraction and the computational burden [38]. There is a complete arc region in the image (Figure 3a), and it has the maximum intensity. The global binarization is first conducted with the maximum intensity (Figure 3b). Then, the region below the bottom boundary of the arc region is the ROI in this work (Figure 3c).

### 2.1. Gabor Filtering

Gabor filtering is a classic orientation feature detection method, which is mostly used in image or video processing [32]. The orientation features of the WSP are salient, and most of them do not change much except for the part of the weld bead during the multipass welding process. A great number of tests show that three specific filtering angles of −12°, 15°, and 90° can effectively highlight the WSP from the arc background. The orientation feature map is calculated as
(1)$$F_{o}=0.2G_{-12{}^{\circ}}+0.3G_{15{}^{\circ}}+0.5G_{90{}^{\circ}}$$
where $F_{o}$ is the computational orientation feature map and *G* represents the Gabor filtering result. The weight 0.5 accounts for the relatively low intensity of the groove region (Figure 3a).

### 2.2. Local Thresholding

The local thresholds are defined as
(2)$$LT_{i}=\underset{j}{\max}[F_{o}(j-2:j+2,i-2:i+2)]$$
where *i* and *j* represent the row and column of the image, respectively, and $LT_{i}$ is the *i*th local threshold used for the region ranging from (*i* − 2) th to (*i* + 2) th rows when the columns are limited from *j* − 2 to *j* + 2.

### 2.3. WSP Location Using SIFT

SIFT is a method for extracting distinctive invariant features from two target images. It can be used to perform reliable matching between different views of an object or scene. This function is invariant to image scale and rotation [39]. In this work, the reference image and raw image are the input images of the SIFT algorithm, and the output of this algorithm is a certain amount of matching points. The ratio of vector angles from the nearest to the second nearest neighbor highly influences the number of the matching points that are used to locate the WSP. Tests show that this number increases with the ratio from 0.9 to 0.99 (Figure 4). However, the higher the ratio, the more fake the matching points. The ratio is set to 0.95 in this work (although more fake matching points exist too, they do not affect the extraction result using the proposed scheme).

In the second processing path, the orientation feature map is first binarized via local thresholds. Then, the nearest neighbor clustering is used to mark the segments of the WSP. Third, the cluster that is the nearest to the matching points is considered as a segment of the WSP. The WSP extraction process is illustrated in Figure 5. Two kinds of arrows represent the two processing paths. The method proposed here adapts to variable WSPs with higher robustness. It is a valuable reference to visual information acquisition, in which visual-sensing-based technologies are used in the automatic welding process.

## 3. Feature Point Identification

The feature points in this work mean the boundary points of the weld bead and the groove. They are usually identified with least squares fitting [36], Hough transform [38], search algorithm [40], slope detection [41], etc. The challenges of identifying all feature points are the variable WSPs and the unknown number of the feature points during the multipass GMAW process. In order to overcome these adverse factors and adapt to possible imperfect WSP extraction results (some interference data points remain), this work presents an effective feature point identification method, as shown in Figure 6.

The extracted WSP is first thinned (the average in the vertical direction) before calculating the slopes. Then, the linear WSP is interpolated with the least square method. The slopes easily fluctuate because of the remaining interference data points and the distorted data points of the WSP. The slope calculation is defined as
(3)$$k_{i}=\frac{1}{n}\sum_{j=1}^{n}\frac{y_{i}-y_{j}}{x_{i}-x_{j}}$$
where n is the number of the involved data points for calculating the *i*th slope, and k is used to represent the slope vector. k still fluctuates abnormally. Without any loss of generality, a one-dimensional linear filter is used to further smooth k with the size 1 × 9. Piecewise polynomial fitting is used to approach the actual variation characteristics of k as
(4)$$Q_{i}=\sum_{j=0}^{30}a_{j}^{i}(k_{h_{i}}{)}^{j}(i=1,2)$$
where $Q_{i}(i=1,2)$ is the fitting slope vector, $a_{j}^{i}(i=1,2)$ are two sets of coefficients, and $k_{h_{i}}$ is each half of k. Monotone interval acquisition is implemented with (5):(5)$$\left\{ \begin{array}{l} k(i-1)\geq k(i)\geq k(i+1)\\ k(i-1)\leq k(i)\leq k(i+1) \end{array}\right..$$

$MI=\left\{mi_{i}\right\},i=1,2,\cdots N$ represents the monotone intervals, and $\left\{{mi}_{i}^{b}\right\}$ and $\left\{{mi}_{i}^{e}\right\}$ are respectively the start and end slopes in MI (Figure 7). The length of each monotone interval is defined as $l_{MI}^{i}=\left|\left\{{mi}_{i}^{b}\right\}-\left\{{mi}_{i}^{e}\right\}\right|$. $l_{MI}^{i}(i=1,2,\cdots )$ are sorted in the ascending order. After the number (e.g., *P*) of the feature points has been supervised (designated), the first *P* monotone intervals of $MI^{\prime }$ are selected. The center of each selected monotone interval indicates the position of the feature points. 

The effectiveness of the feature point identification method proposed here is validated using some continuous sampling images shown in Figure 8. The experimental results show that the feature points of 95.6% of the images can be identified precisely with the error of 3 pixels. The feature points deviate from the actual positions when there is some interference such as splash, welding slag, etc., near these positions (images in red rectangular in Figure 8). An error correction mechanism is necessary to predict/optimize the feature point when ineffective identification results occur. Multithreading processing is used to predict/optimize all feature points that are utilized to implement the measurement of WBGFs.

## 4. Cubic Exponential Smoothing for Stabilizing Feature Point Identification Process

The cubic exponential smoothing method is typically used to stabilize time serial variables through predicting the real state for the next several sampling periods. This method is effective when the time serial variable fluctuates periodically in a linear trend. The identified feature points have this characteristic when the welding torch moves from the start to the end welding positions. Thus, they can be predicted with the cubic exponential smoothing method when the ineffective feature point identification result is diagnosed. The state of the target feature point is iterated as
(6)$$\left\{ \begin{array}{l} S_{t}^{\left(1\right)}=\alpha x_{t}+(1-\alpha )S_{t-1}^{\left(1\right)}\\ S_{t}^{\left(2\right)}=\alpha S_{t}^{\left(1\right)}+(1-\alpha )S_{t-1}^{\left(2\right)}\\ S_{t}^{\left(3\right)}=\alpha S_{t}^{\left(2\right)}+(1-\alpha )S_{t-1}^{\left(3\right)} \end{array}\right.$$
where α is the smoothing coefficient, and $x_{t}$ is the coordinate of the target feature point in the *y*-/*x*-direction in images (suppose that they are independent). The coordinate of the target feature point is predicted/optimized as
(7)$$x_{t+T}=A_{t}+B_{t}T+C_{t}T^{2}$$
where *T* is the sampling period, and the coefficients *A*, *B* and *C* are defined as
(8)$$\left\{ \begin{array}{l} A_{t}=3S_{t}^{\left(1\right)}-3S_{t}^{\left(2\right)}+S_{t}^{\left(3\right)}\\ B_{t}=\frac{\alpha }{2{\left(1-\alpha \right)}^{2}}[(6-5\alpha )S_{t}^{\left(1\right)}-2(5-4\alpha )S_{t}^{\left(2\right)}+(4-3\alpha )S_{t}^{\left(3\right)}]\\ C_{t}=\frac{\alpha^{2}}{21{\left(1-\alpha \right)}^{2}}[S_{t}^{\left(1\right)}-2S_{t}^{\left(2\right)}+S_{t}^{\left(3\right)}] \end{array}\right.$$
where $x_{t}$ is initialized with the coordinate of the corresponding feature point of the reference image, namely the designated tracking position.

The trigger of starting the smoothing process is that the designated tracking position deviates from its last position by 3 pixels. One-step prediction is used in this work. The typical prediction process is given in Algorithm 1.
**Algorithm 1** Feature point optimization process using cubic exponential smoothing.Input $x_{t},\alpha$
Output $x_{t+1}$
• Feature point ineffectiveness judgement: $\left\|x_{t+1}^{\prime }-x_{t}\right\|\geq 3pixels$  *****
$S_{t}^{\left(1\right)}=\alpha x_{t}+(1-\alpha )S_{t-1}^{\left(1\right)}$   $S_{t}^{\left(2\right)}=\alpha S_{t}^{\left(1\right)}+(1-\alpha )S_{t-1}^{\left(2\right)}$   $S_{t}^{\left(3\right)}=\alpha S_{t}^{\left(2\right)}+(1-\alpha )S_{t-1}^{\left(3\right)}$  ***** Calculate $A_{t}$, $B_{t}$ and $C_{t}$   $A_{t}=3S_{t}^{\left(1\right)}-3S_{t}^{\left(2\right)}+S_{t}^{\left(3\right)}$   $B_{t}=\frac{\alpha }{2{\left(1-\alpha \right)}^{2}}[(6-5\alpha )S_{t}^{\left(1\right)}-2(5-4\alpha )S_{t}^{\left(2\right)}+(4-3\alpha )S_{t}^{\left(3\right)}]$   $C_{t}=\frac{\alpha^{2}}{21{\left(1-\alpha \right)}^{2}}[S_{t}^{\left(1\right)}-2S_{t}^{\left(2\right)}+S_{t}^{\left(3\right)}]$  ***** Calculate $x_{t+1}$   $x_{t+1}=A_{t}+B_{t}+C_{t}$
• $x_{t+1}^{\prime }$ is replaced with $x_{t+1}$  $x_{t+1}^{\prime }$ is the coordinate of the current identified feature point.

The designated tracking position in the images within the red rectangle in Figure 8 is optimized with the cubic exponential smoothing method shown in Figure 9. The optimized data in the y-/x-direction are given in Figure 10. The accuracy of identifying the feature points is within 1.50 pixels through this diagnostic process overcoming any random interference. 

Experimental results show that the cubic exponential smoothing method enhances the accuracy of feature point identification. It can be used to restrain the abnormal fluctuation of the tracking position in visual-sensing-based automated GMAW. This process stabilizes seam tracking as well as the subsequent WBGF modeling.

## 5. Modeling WBGFs

### 5.1. Sub Pixel Discrimination of WBGFs

In order to improve modeling accuracy, linear interpolation is applied to sub pixel discrimination of the weld bead before modeling as
(9)$$O_{i}^{\prime }=\frac{A_{i+1}-A_{i}^{\prime }}{A_{i+1}-A_{i}}O_{i}+\frac{A_{i}^{\prime }-A_{j}}{A_{i+1}-A_{i}}O_{i+1}$$
where $O_{i}$, $A_{i}$, $O_{i+1}$, and $A_{i+1}$ are the coordinates of two adjacent pixels in the region of the weld bead, and $A_{i}^{\prime }$ and $O_{i}^{\prime }$ are the coordinates of interpolation points. Nine points are linearly interpolated between two adjacent pixels in this work.

### 5.2. Modeling Process of WBGFs

The reference profile (Figure 11) is from the reference image. The two profiles are aligned by overlapping a common feature point (e.g., the leftmost feature point). The bead region is determined after some feature points of two input WSPs are designated respectively. The region consists of the lower and upper curve lines. That is, the upper boundary is first determined with the corresponding feature points, and then the start point of the lower boundary is the nearest point to the left of the upper boundary. The end point of the lower boundary is also the nearest point to the right of the upper boundary. There are two gaps between the two boundaries. Two lines are defined with the two-point form of straight-line equations (Figure 12) to connect the two boundaries:(10)$$y^{i}=\frac{y_{1i}-y_{2i}}{x_{1i}-x_{2i}}(x-x_{1i})+y_{1i}(i=1,2)$$
where $x_{1i}$, $y_{1i}$, $x_{2i}$, and $y_{2i}(i=1,2)$ are the coordinates of the two sets of endpoints of the curve lines. Thus, the weld bead in the images is a closed region.

$\left\{Dt_{i}^{1}\right\}(i=1,2,\ldots Q)$ and $\left\{Dt_{i}^{2}\right\}(i=1,2,\ldots M)$ are used to represent the data points respectively belonging to the upper and lower boundaries. Polynomial fitting is used to model the two boundaries as
(11)$$\left\{ \begin{array}{l} f_{1}(Dt_{i}^{1},W_{1})=\sum_{j=0}^{N}\omega_{j}{\left(Dt_{i}^{1}\right)}^{j}\\ f_{2}(Dt_{i}^{2},W_{2})=\sum_{j=0}^{N}\omega_{j}^{\prime }{\left(Dt_{i}^{2}\right)}^{j} \end{array}\right.$$
where N is the order and initialized to 10. In order to implement the all-position measurement, two ranges of the weld bead are first determined as $\left\{LR_{j}\right\}(\left\{LR_{j}\right\}={\left\{f_{1}(Dt_{i}^{1},W_{1})\right\}}_{x}\cap {\left\{f_{2}(Dt_{i}^{2},W_{2})\right\}}_{x})$ and $\left\{UD_{j}\right\}(\left\{UD_{j}\right\}={\left\{f_{1}(Dt_{i}^{1},W_{1})\right\}}_{y}\cup {\left\{f_{2}(Dt_{i}^{2},W_{2})\right\}}_{y})$ in the y-/x-direction, respectively. For $\forall a_{j}\in [\min(\left\{LR_{j}\right\},\max(\left\{LR_{j}\right\}))]$, there exist two intersection points between the line $x=a_{j}$ and the closed weld bead region. Their ordinates are $A_{j1}$ and $A_{j2}$ in the y-direction, respectively. In addition, for $\forall b_{j}\in [\min(\left\{UD_{j}\right\},\max(\left\{UD_{j}\right\}))]$, there also exist two intersection points between the line $y=b_{j}$ and the bead region. The two abscissas are $O_{j1}$ and $O_{j2}$ in the x-direction, respectively (Figure 13). Thus, the *APWBW* is then modeled with 12, the *APWBH* is modeled with 13, and the area is acquired with 14.
(12)$$APWBW={\left|O_{j1}-O_{j2}\right|}_{j}$$
(13)$$APWBH={\left|A_{j1}-A_{j2}\right|}_{j}$$
(14)$$Area=(\int \left[f_{1}(Dt_{i}^{1},W_{1}\right)-f_{2}(Dt_{i}^{2},W_{2})]dx|{}_{\min(\left\{LR_{j}\right\})}^{\max(\left\{LR_{j}\right\})}$$

In addition, the zero and first moment of the binary image are used to calculate the coordinate of the center of the gravity of the weld bead as
(15)$$\left\{ \begin{array}{l} x_{c}=\sum_{i}\sum_{j}j\cdot v(i,j)/\sum_{i}\sum_{j}v(i,j)\\ y_{c}=\sum_{i}\sum_{j}i\cdot v(i,j)/\sum_{i}\sum_{j}v(i,j) \end{array}\right.$$
where v(i,j) is the gray value at point (i,j), and $x_{c}$, $y_{c}$ are the corresponding coordinates of the center of gravity. It is marked with “+” in images in the subsequent modeling experiments.

## 6. Experimental Results

The welding system is shown in Figure 14. The modeling processes with T-joints of 30 mm and 50 mm thickness are first used to show the effectiveness of the method proposed in this work (Figure 15, Figure 16, Figure 17 and Figure 18). *APWBW* modeling is conducted from top to bottom. *APWBH* modeling is carried out from left to right. The feature point optimization process based on the cubic exponential smoothing method enhances the modeling accuracy (Figure 16 and Figure 18). 

The robustness of the modeling method proposed in this work is further investigated with the variable WSP by changing the welding current during the same welding process (Figure 17 and Figure 18).

Two modeling processes with a butt joint of 30 mm thickness are conducted to further show the effectiveness of the proposed method here (Figure 19 and Figure 20). These experimental results show that the proposed method in this work still meets the modeling requirement in butt-joint multipass GMAW regarding these typical WBGFs.

The modeling result on the lap-joint weld bead (Figure 21, Figure 22 and Figure 23) is also used to show the effectiveness of the proposed method here. 

These modeling experiments show that this method can be applied to the real-time modeling of the WBGFs with typical joints, thin or thick steel plates. 

## 7. Discussion

This paper presented an effective WBGF modeling method during the multipass GAMW process with T-joints and butt joints based on machine vision and learning. This method can stably acquire the area, center of gravity, and *APWBW* and *APWBH* of the weld bead in real time. This study expanded the traditional feature measurement of irregular weld beads. It contributes to weld formation control and planning by providing more evidence to optimize the welding process parameters. In addition, the improvements on WBGF modeling in this work include an attempt at real-time modeling, the stable modeling process, and the adaptability of the variable modeling objects.

This work proposed an “all-position” concept. The all-position width and height of the weld bead contain more useful information to optimize the welding process parameter compared with [24,25]. In addition, the imaging of the welding wire is worthy of research attention, because it can provide direct evidence to control the posture of the welding torch.

The modeling processes show that the one-dimensional filtering size (from 1 × 3 to 1 × 29) is the most sensitive to the modeling results. It is necessary to develop an adaptive mechanism to approach the appropriate setting, which is the first aspect to be improved in the future study. Polynomial fitting is currently the popular method to model the objects. However, there is an over-fitting phenomenon. Although this work uses piecewise fitting to respectively represent the upper and lower boundaries of the weld bead, over-fitting still happens. For example, another mechanism monitoring the maximum error between the last several data points of the actual boundary and the fitting result should be built to optimize the order of the polynomial function. The relationship between the visual features of the weld bead and welding process parameters will be investigated in the next study.

## 8. Conclusions

This work implemented real-time modeling on the area, center of gravity, and all-position width and height of the weld bead in thick plate gas metal arc welding with T-joints and butt joints based on visual sensing. Some conclusions about visual information acquisition, fault detection and diagnosis, and the weld bead geometry feature modeling method are given as follows.
(1)The proposed feature point identification method combined with the weld seam profile extraction method adapts to the various weld seam profiles. It provides the valuable reference to visual information acquisition for visual-sensing-based automated welding.(2)The proposed fault detection and diagnosis of feature point identification based on the cubic exponential smoothing method shows that this optimization process enhances the identification accuracy to 1.50 pixels. This method shows its potential application value for improving tracking accuracy and welding quality.(3)The proposed modeling method in this work can obtain the area, center of gravity, and all-position width and height of the weld bead in real time in gas metal arc welding with typical joints. This modeling method provides more effective evidence to control the weld formation and planning, particularly during the multipass arc welding process.

## Figures and Tables

**Figure 1 sensors-20-07104-f001:**
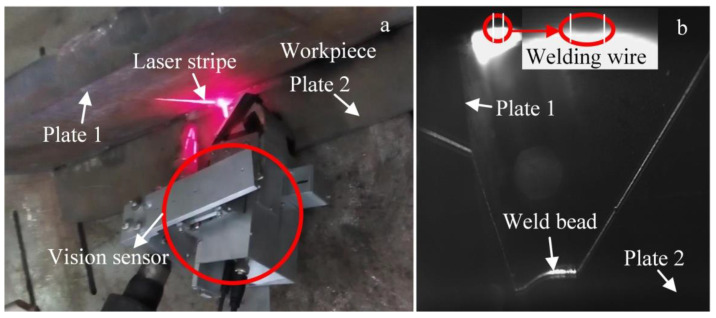
(**a**) Vision sensor. (**b**) Typical image captured during the welding process.

**Figure 2 sensors-20-07104-f002:**
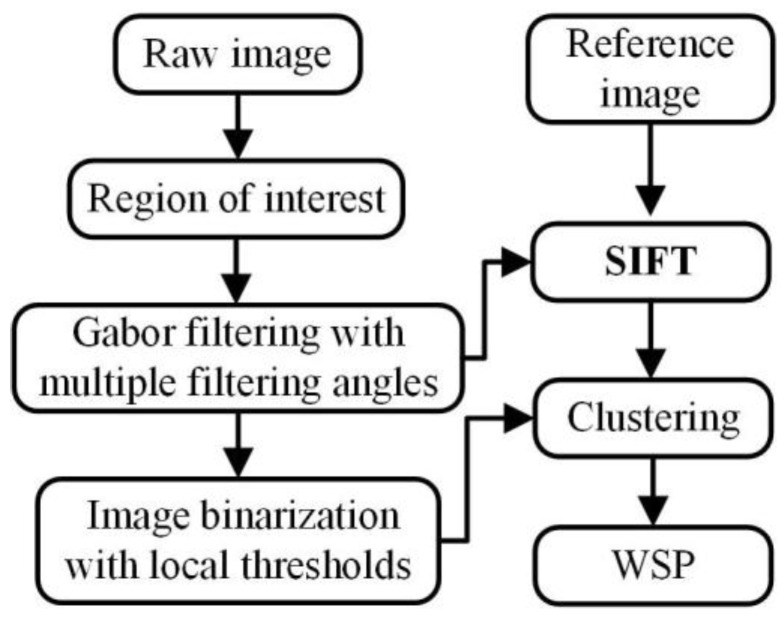
Flow chart of the weld seam profiles (WSP) extraction method proposed in this work.

**Figure 3 sensors-20-07104-f003:**
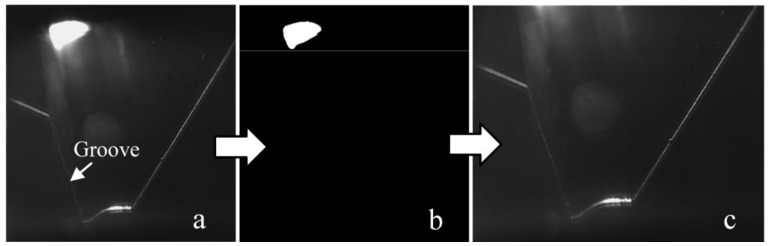
Region of interest (ROI) determination process. (**a**) Raw image. (**b**) Binarization result to determine the ROI. (**c**) ROI.

**Figure 4 sensors-20-07104-f004:**
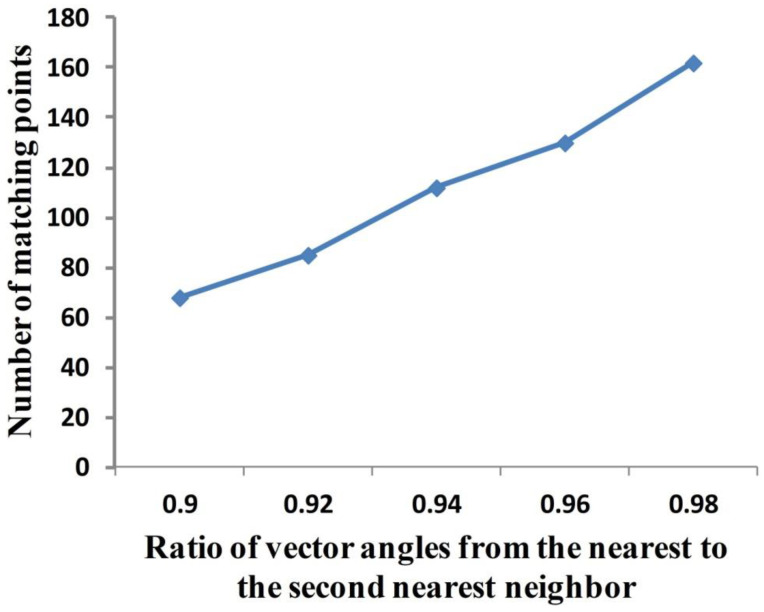
Relationship between the ratio of vector angles from the nearest to the second nearest neighbor and the number of matching points.

**Figure 5 sensors-20-07104-f005:**
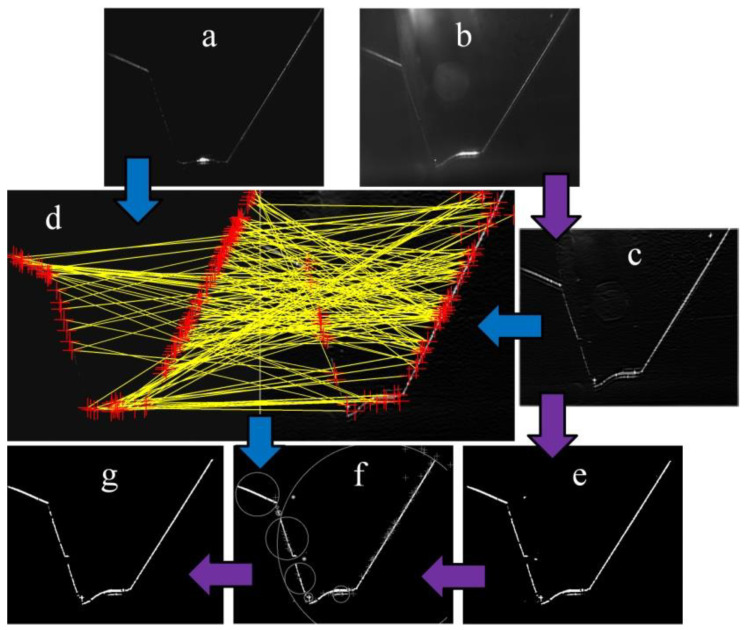
WSP extraction process based on scale-invariant feature transform (SIFT) and clustering. (**a**) Reference image. (**b**) ROI. (**c**) Orientation feature map. (**d**) Feature matching result using SIFT (red “+” represents the matching points). (**e**) Binarization result using local thresholds. (**f**) Clustering result. (**g**) WSP extraction result.

**Figure 6 sensors-20-07104-f006:**
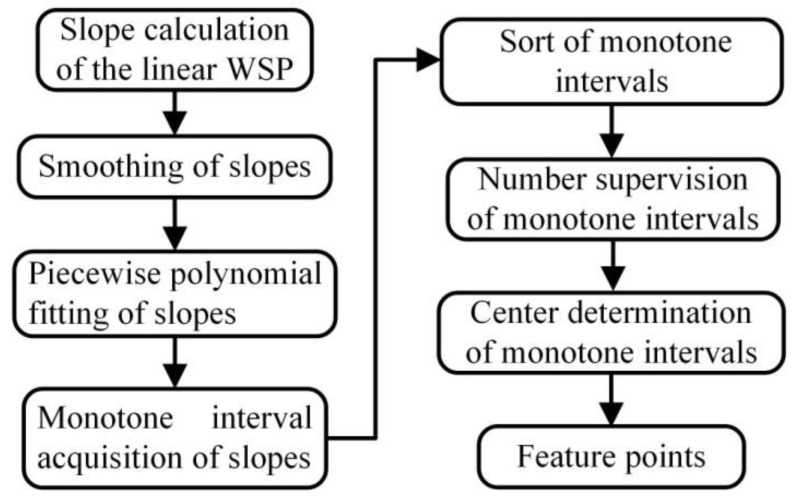
Flow chart of identifying the feature points.

**Figure 7 sensors-20-07104-f007:**
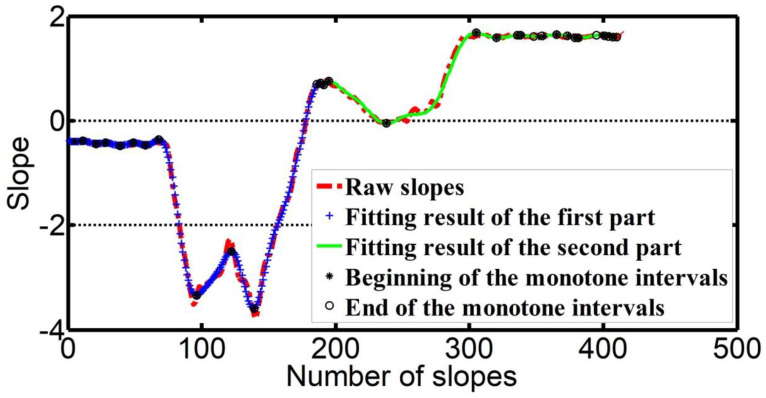
Slope piecewise fitting and monotone interval acquisition of slopes.

**Figure 8 sensors-20-07104-f008:**
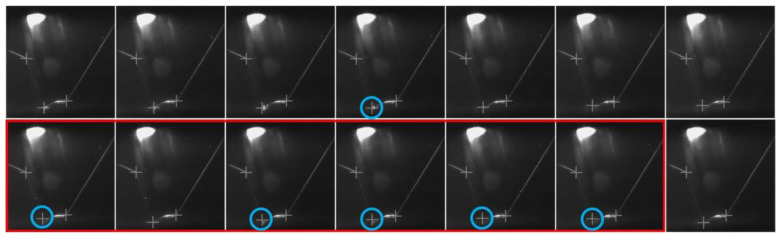
Feature point identification results in which the images in the red rectangle show the examples of ineffective feature identification because of little welding slag (blue circles mark the ineffective feature points); the feature point in the middle is the tracking position.

**Figure 9 sensors-20-07104-f009:**
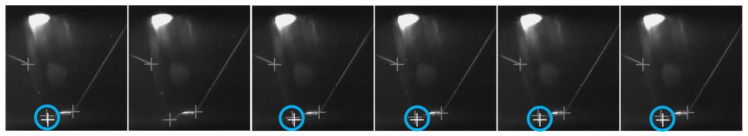
Example of optimizing the designated feature point using the cubic exponential smoothing method (blue circle indicates the tracking position, and white “+” is the optimized result).

**Figure 10 sensors-20-07104-f010:**
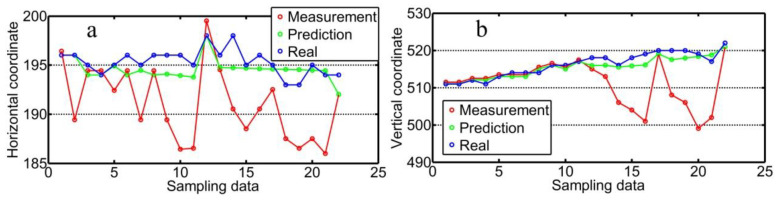
(**a**) Optimization result of the designated tracking position in the x-direction with an accuracy of 1.56 pixels and (**b**) in the y-direction with an accuracy of 1.43 pixels.

**Figure 11 sensors-20-07104-f011:**
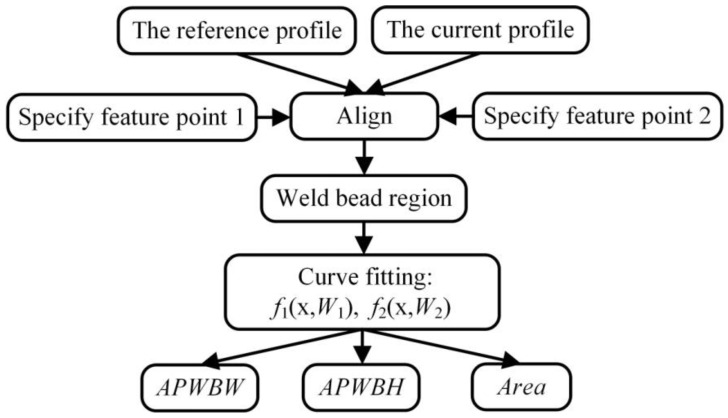
Scheme of modeling weld bead geometry features (WBGFs).

**Figure 12 sensors-20-07104-f012:**
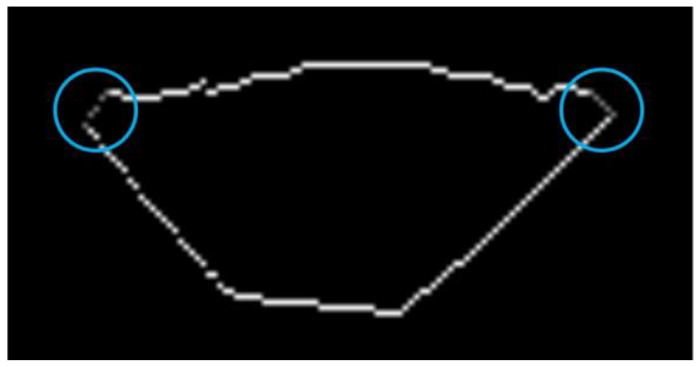
Example of filling the gaps between two boundaries of the weld bead (two blue circles indicate the gaps that are filled with the gray data points).

**Figure 13 sensors-20-07104-f013:**
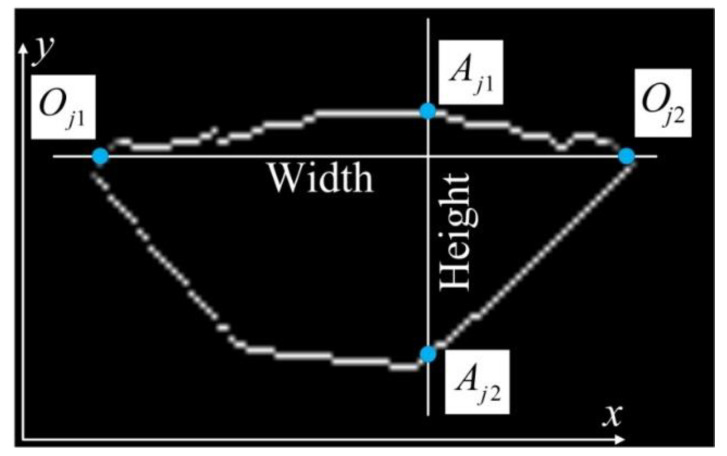
Diagrammatic sketch of determining the *APWBW* and *APWBH*.

**Figure 14 sensors-20-07104-f014:**
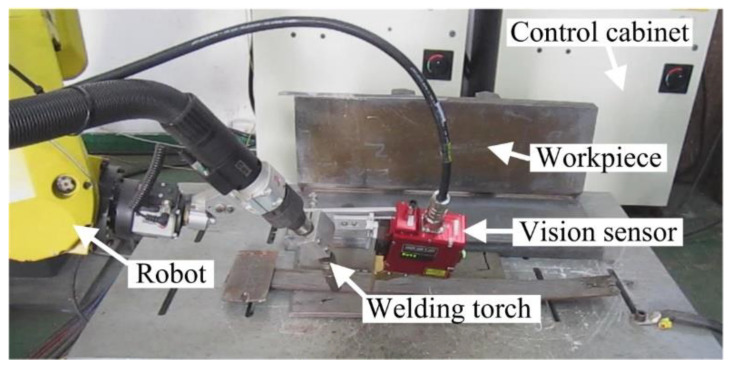
Welding workstation.

**Figure 15 sensors-20-07104-f015:**
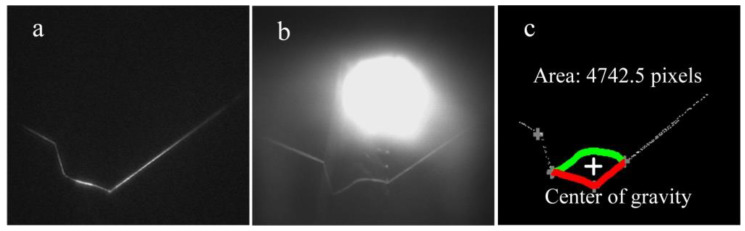
WBGF modeling result with (**a**) T-joint of 30 mm thickness. A reference image. (**b**) Current WSP. (**c**) Region and area of the weld bead (“+” makes the center of gravity). (**d**) Modeling result of the *APWBW* and (**e**) of the *APWBH*. (**f**) 3D reconstruction of the weld bead.

**Figure 16 sensors-20-07104-f016:**
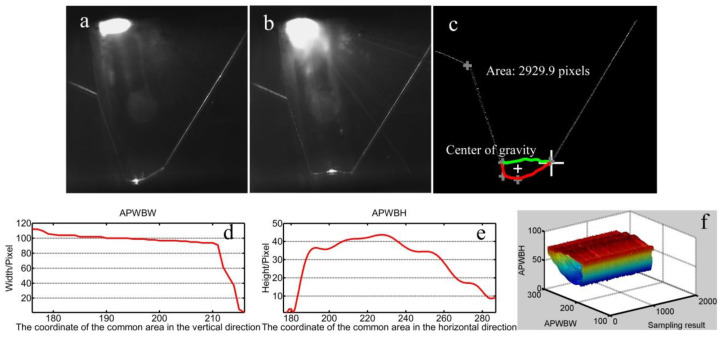
WBGF modeling result with a T-joint of 50 mm thickness. (**a**) Reference image. (**b**) Current WSP. (**c**) Region and area of the weld bead (bigger white “+” represents the optimization position with the cubic exponential smoothing method; similarly hereinafter). (**d**) Modeling result of the *APWBW* and (**e**) of the *APWBH*. (**f**) 3D reconstruction of the weld bead.

**Figure 17 sensors-20-07104-f017:**
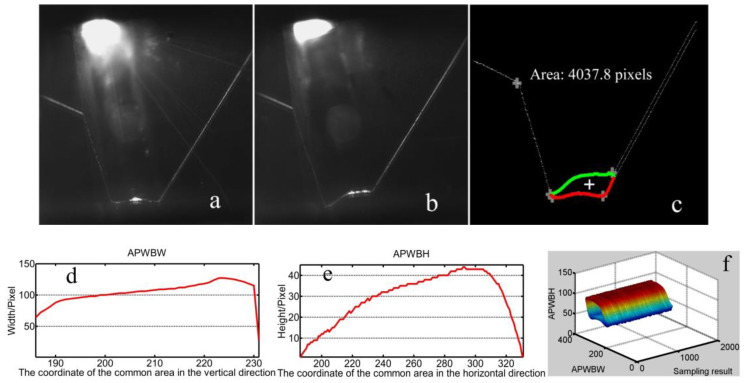
(**a**) Reference WSP. (**b**) Current WSP with bigger welding current. (**c**) Modeling results of the area and center of gravity. (**d**) Modeling result of the *APWBW* and (**e**) of the *APWBH*. (**f**) 3D reconstruction of the weld bead.

**Figure 18 sensors-20-07104-f018:**
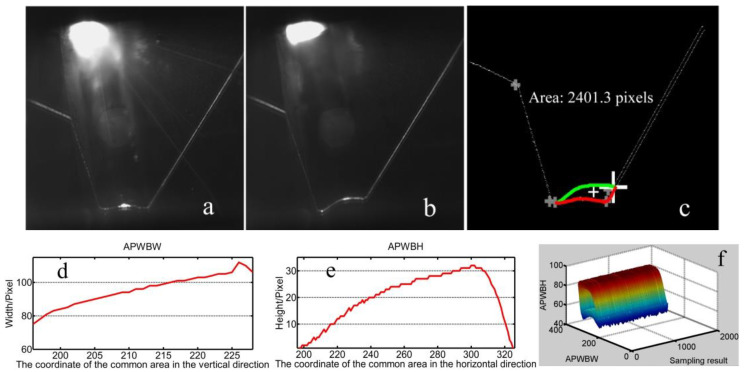
(**a**) Reference WSP. (**b**) Current WSP with smaller welding current (reduce from about 210 A to 170 A). (**c**) Modeling results of the area and center of gravity. (**d**) Modeling result of the *APWBW* and (**e**) of the *APWBH*. (**f**) 3D reconstruction of the weld bead.

**Figure 19 sensors-20-07104-f019:**
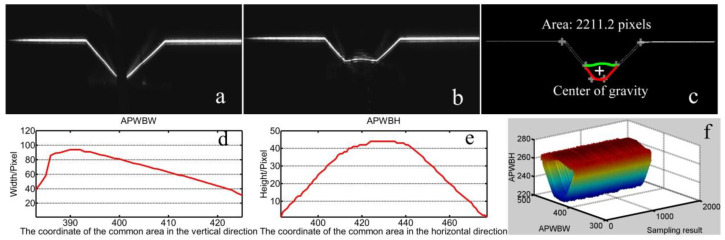
Modeling results with a butt joint to show the effectiveness of the proposed method here. (**a**) Reference WSP. (**b**) Current WSP. (**c**) Modeling results of the area and center of gravity. (**d**) Modeling result of the *APWBW* and **(e****)** of the *APWBH*. (**f**) 3D reconstruction of the weld bead.

**Figure 20 sensors-20-07104-f020:**
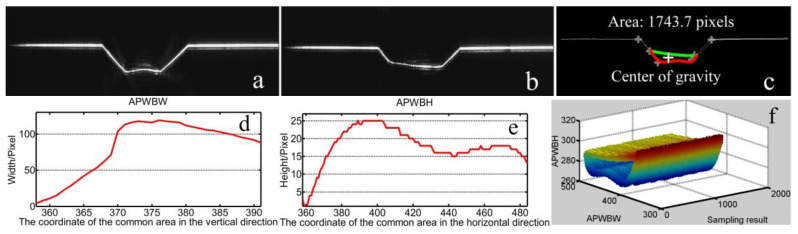
(**a**) Reference WSP. (**b**) Current WSP. Modeling results of the area and center of gravity. (**c**) Modeling results of the area and center of gravity. (**d**) Modeling result of the *APWBW* and (**e**) of the *APWBH*. (**f**) 3D reconstruction of the weld bead.

**Figure 21 sensors-20-07104-f021:**
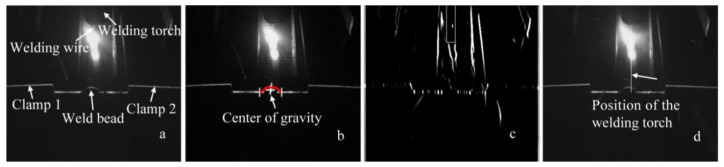
(**a**) Raw image with a lap joint. (**b**) Modeling result. (**c**) Orientation feature of the welding wire. (**d**) Position mark of the welding wire.

**Figure 22 sensors-20-07104-f022:**
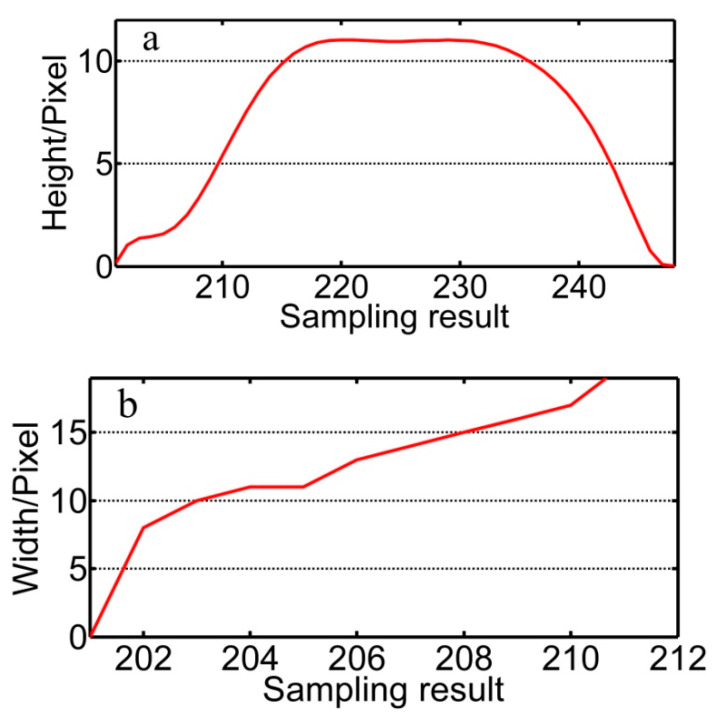
(**a**) All-position height and (**b**) width.

**Figure 23 sensors-20-07104-f023:**
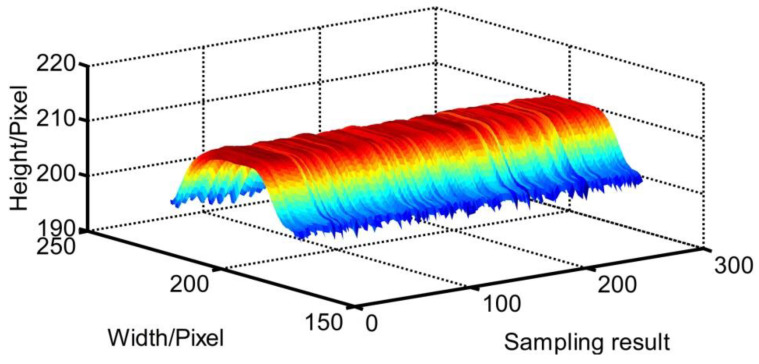
Three-dimensional reconstruction of the weld bead.

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
