# Peer review of "Dynamic Modeling of Weld Bead Geometry Features in Thick Plate GMAW Based on Machine Vision and Learning"

_sensors, 2020, doi:10.3390/s20247104_

Round 1

Reviewer 1 Report

This work presents a method for analyzing the geometry of the weld bead in GMAW welding. The article is interesting and details the necessary mathematical treatment quite well. A practical application is also presented, a fact that strengthens the study.

Small faults to be corrected can be found in the attached document.

Author Response

Thanks for the great suggestions!

1.Reference image

"Reference image is the one captured before arc starting". It is presented in the 129th line. Actually, the reference image can provide lots of useful information to locate the laser stripe that is in the image captured during the welding process.

2. format in Line 139.

Thanks a lot. Our mistake. We have revised it.

3. Bizarization?

Sorry. It should be Binarization, and we have revised it.

Thanks again.

Reviewer 2 Report

The manuscript entitled ‘Dynamic modeling of weld bead geometry features in thick plate GMAW manufacturing with typical joints based on machine vision and learning’ falls within the scope of the journal Sensors. The paper contains very interesting experimental results as well as measurement procedures and algoritms. It is of sufficient scientific interest and has originality in its technical content to merit publication. The authors have cited the relevant literature. Methods, interpretations of results are correct. The authors presented extensive material supporting the conducted research. The issues were well presented. In terms of content, the analysis does not raise any objections. The arrangement of work maintains substantive continuity and constitutes a logical whole. The conclusions are correct, new and well-worded. It is worth emphasizing that the research results do not matter only the cognitive ability but can be practically used in the modeling of weld bead geometry. The manuscript is suitable for publication in its present form.

Author Response

Thanks a lot for the comment.

"The manuscript entitled ‘Dynamic modeling of weld bead geometry features in thick plate GMAW manufacturing with typical joints based on machine vision and learning’ falls within the scope of the journal Sensors. The paper contains very interesting experimental results as well as measurement procedures and algoritms. It is of sufficient scientific interest and has originality in its technical content to merit publication. The authors have cited the relevant literature. Methods, interpretations of results are correct. The authors presented extensive material supporting the conducted research. The issues were well presented. In terms of content, the analysis does not raise any objections. The arrangement of work maintains substantive continuity and constitutes a logical whole. The conclusions are correct, new and well-worded. It is worth emphasizing that the research results do not matter only the cognitive ability but can be practically used in the modeling of weld bead geometry. The manuscript is suitable for publication in its present form."

We will carefully check the paper again.

Reviewer 3 Report

This paper is mainly described that the theory of a new effective WBGF modeling method related to real operation. There indeed is much work on this research. The authors properly applied lots of functions to establish models with higher accuracy. There is much description of models and experiments, but there is little analysis of them. And some statements are not proper but difficult to understand. Some sentences are even confused. Also I think that section 1. Indtoduction should be better. A lot of important papers not analized. For example: Henckell, P., Gierth, M., Ali, Y., Reimann, J., Bergmann, J.P. Reduction of energy input in wire arc additive manufacturing (WAAM) with gas metal arc welding (GMAW) (2020) Materials, 13 (11), paper â„– 2491. DOI: 10.3390/ma13112491; Chen, X., Su, C., Wang, Y., Siddiquee, A.N., Konovalov, S., Jayalakshmi, S., Singh, R.A. Cold Metal Transfer (CMT) Based Wire and Arc Additive Manufacture (WAAM) System (2018) Journal of Surface Investigation, 12 (6), pp. 1278-1284.  DOI: 10.1134/S102745101901004X;  Liu, K., Chen, X., Shen, Q., Pan, Z., Singh, R.A., Jayalakshmi, S., Konovalov, S. Microstructural evolution and mechanical properties of deep cryogenic treated Cu–Al–Si alloy fabricated by Cold Metal Transfer (CMT) process (2020) Materials Characterization, 159, paper â„– 110011.  DOI: 10.1016/j.matchar.2019.110011.

It is good paper but need make better.

Author Response

Thanks a lot for the wonderful suggestions.

1. There is much description of models and experiments, but there is little analysis of them.

Thanks. We have added some analysis on the modeling method such as Line 361These modeling experiments show that this method can be applied to the real-time modeling of the WBGFs with typical joints, thin or thick steel plates.”, and Line 372 “This work proposed a concept “all-position”. The all-position width and height of the weld bead contain more useful information to optimize the welding process parameter compared with [24, 25]. In addition, the imaging of the welding wire is worthy of research concentration because it can provide direct evidence to control the posture of the welding torch.

2. And some statements are not proper but difficult to understand. Some sentences are even confused. 

This is true. We have revised the article sentence by sentence, and marked these revised sentences with red. Thanks.

3. Also I think that section 1. Indtoduction should be better. A lot of important papers not analized.

We reorganized Section 1, and think that the study on wire arc additive manufacturing and accurate metal deposition control using Cold Metal Transfer (CMT) process should be actually analized in the introduction section. Therefore, we added the papers mentioned

In article:

1. there is a non-proper statement in the number of 20 and 21 sentence. Something is proposed if adding "with" that means that something will be proposed along previous one.

Thanks. We have punctuated the sentence, and "with" is replaced with "using".

2. "Before modeling of WBGFs, " is not proper......

We revised it as "A linear interpolation method is presented to implement sub pixel discrimination of the weld bead before modeling WBGFs".

3. “engineering equipment and pipe manufacturing [1, 2] etc.”  the statement is not correct.

Thanks. We revised it as "engineering equipment production, pipe manufacturing [1, 2] etc."

4. "Weld bead geometry features (WBGFs) can reflect welding process parameters and provide effective evidences to suggest how to optimize welding process parameters because the relationship can be built with various models [8-10]." it is confused to understand the thought of this sentence.

We rewrote the sentence as "Weld bead geometry features (WBGFs) can reflect welding process parameters and provide effective evidences to suggest how to optimize the latter because the relationship between the two aspects can be built with various models [8-10]". Thanks.

5. "For the active vision system, researchers generally use the laser to illuminate the weld bead to create a deformation proportional to the bead width and height." the statement is not proper.

We rewrote the sentence as "For the active vision system, researchers generally use the laser to illuminate the weld bead to create a deformation that is proportional to the bead width and height.".

6. Paragraph 5 might be related to the last second one. the situation of this paragraph is not good. 

Thanks. We intend to use this paragraph to summarize the current study status of modeling WBGFs. We still think the situation of it is proper when a sentence is added at the beginning of the next paragraph: WSP extraction must be implemented during the active-sensing-based WBGF modeling process.

7. "The WSP extraction method is thus proposed based on SIFT and machine learning shown in Fig. 2."  thus is better used in the beginning of sentence.

Thanks a lot. We revised the sentence.

8."raw image means the image captured during the welding process while reference image is the one captured before arcing." The statement is confused.

We revised the sentence as "raw image means the image captured during the welding process while reference image is the one captured before arc starting (the welding torch is at the welding position but the electric welding machine does not start to work yet)." We hope that this statement is clear.

9. “However, the higher the ratio, the more are fake matching points. The ratio is set to 0.95 in this work.” if it is more fake with higher ratio, why the ratio is set to 0.95?

We added explanations as "The ratio is set to 0.95 in this work to locate more segments of the WSP (although more fake matching points exist too, they do not affect the extraction result using the proposed scheme)." In addition, this sentence was moved to the next paragraph.

10. "The two kinds of arrows represent two processing paths. " there is little description about two processing paths.

In fact, we described the first processing path from Line 130. The description about the second processing path is from Line 173.

11. "Due to the discrete data points of the extracted WSP, the slopes of the extracted WSP easily fluctuate with the remaining interference data points and the distorted data points after the extracted WSP has been thinned (the average in the vertical direction) and interpolated with the least square method. " it is difficult to understand. there is much information in one sentence. it is better divided into two sentence logically.

Thanks. The long sentence was reorganized as "Before calculating the slopes, the extracted WSP is thinned (the average in the vertical direction). Then the linear WSP is interpolated with the least square method. Due to the discrete data points of the linear WSP, the slopes of the linear WSP easily fluctuate with the remaining interference data points and the distorted data points."

12. “the start point of the lower boundary is then the nearest.....", it should be put at the beginning of this half-sentence.

We rewrote it as "then the start point of the lower boundary is the nearest".

13. "In addition, we think that the center of gravity of the weld bead is also a useful feature, which can guide the effective adjustment of the angle of the welding torch." It is better not use the first-person perspective in research.

Thanks . We deleted "we think that".

14. "We investigate the robustness of the measurement method...."

Thanks. We rewrote the sentence as "The robustness of the measurement method proposed in this work is investigated......".

15. "We also use the modeling result on the lap-joint weld bead (Figs. 22-24) to...."

We rewrote the sentence as "The modeling result on the lap-joint weld bead (Figs. 22-24) is also used to...."

Reviewer 4 Report

It is an interesting study about weld bead modeling, congratulations.

The title is very long, please shorten it.

General remarks:

However the reviewer is not a native English speaking person, the language needs serious improvement, there are sentences very hard to understand.

Figures need improvement, also their captions, there are in some cases not very informative. E.g. I couldn't tell the difference between the images of Fig.8 or fig 9. are they all necessary?

Welding parameters are not presented, it should be in the manuscript.

I couldn't find the validation of the method, were there any measurements taken e.g. on metallographic cross section to veryfi the model?

Conclusions is way too short for a long paper like this, and it should be self-contained a lot of colleagues read it first , than the paper (sometimes me too)

I made my specific remarks, questions, comments in the manuscript_with_reviewers_comments

With proper corrections I think the manuscript could satisfy the publication criteria in Sensors.

Author Response

Thanks a lot!

1.The title is very long, please shorten it.

Thanks. We shorten it as “Dynamic modeling of weld bead geometry features in thick plate GMAW based on machine vision and learning”.

2. The language needs serious improvement, there are sentences very hard to understand.

This is true. We have revised the article sentence by sentence, and marked these revised sentences with red. Thanks.

3. Figures need improvement, also their captions, there are in some cases not very informative.

We have revised figures and their captions (from Fig. 8 to Fig. 21) from the context and deleted one.

4. I couldn't tell the difference between the images of Fig.8 or fig 9. are they all necessary?

Actually we have deleted some of them and deleted the previous Fig. 8. The statement “The effectiveness of the feature point identification method proposed here is validated using some continuous sampling images shown in Fig. 8.” has been presented in the revised version to explain why some images are used.

5. Welding parameters are not presented, it should be in the manuscript.

We add the description on welding parameters “reduce from about 210 A to 170 A”, which is added in Fig. 18. As the experiments are so many, we actually omitted this part. However, if the editor demands this content, we can add some tables. Thanks.

6. I couldn't find the validation of the method, were there any measurements taken e.g. on metallographic cross section to veryfi the model?

In our opinion, the modeling processes with different weld beads are validation, such as Figs. 18-20. Actually we just used different weld formation shapes to validate the effectiveness of the proposed modeling method in this work.

7. Conclusions is way too short for a long paper like this, and it should be self-contained a lot of colleagues read it first , than the paper (sometimes me too)

Thanks. We rewrote the conclusion section. This paper is relatively long; we originally intended to write this part simply. However, we added discussion section, and some analysis is placed in this section.

In article:

1. “in an offline and online fashion”. how is online modeling, with physical simulation? 

We think that the modeling process based on visual sensing and image processing is an online fashion because this fashion can obtain the related features of the weld bead online. This statement is given in the next paragraph. 

2. "For the offline modeling process, a better option is the non-contact type measurement methods such as ultrasonic sensing [17] and infrared....."  its in contact mode, but the accuracy is doubtful...

ThanksWe are wrong. This sentence is revised as “The offline modeling process includes ultrasonic sensing [17] and infrared.....”

3. "especially the center of gravity of the weld bead...."  mid line?

In this paper, "center of gravity" is a coordinate. 

4. “special combination of dimmer 110 glass and the filter (Fig. 1).”  there s no mention it on fig 1.

Thanks a lot. In the revised manuscript, we added the related description as "Fig. 1a shows the typical work state of the sensor, and Fig. 1b gives the imaging effect of the laser stripe and the wire".

5. "Experiments show that.....", which experiments? 

Actually, these images presented in this paper are from offline experiments. In the revised manuscript, we used "Experimental results show that ...". Maybe the imaging of the welding wire in Fig. 1 is vague, but Fig. 22 show a clear imaging, and it is an approximate straight line. We use a line to mark it. We added two images to show this content (Fig. 22c and d).

6. unify font sizes! Plate 2?

We unified the font size in this image.

7. whats what on the fig?

Our mistakes. We revised the related captions and added some. Thanks.

8."Two facts are that the image without arc is captured ahead of the move of the welding torch, and this image has the similar laser stripe (the shape, the position and the intensity) with the one captured during the welding process." please rephrase!

Thanks. we rephrased it as "The image that is captured before arc starting has a clear WSP and the similar laser stripe (the shape, the position and the intensity) with the one captured during the welding process."

9. abbrev. not defined yet,

We replaced ROI in Fig. 2 with "Region of interest", and corrected the wrong spelling mistakes.

10. "arcing", welding terminology please

We replaced it with arc starting. It means that the electric welding machine starts to work.

11. "Gabor filter", whats that?

Gabor filter is a kind of filter that is usually applied to object detection in image processing using orientation features. Therefore, it is necessary to set filtering angles.

12. "Binarization result to determine the ROI. c ROI.". different font size, why?

Sorry. Our mistake. It was revised.

13. more informative captions please!

Thanks. We revised the caption of Fig. 4 as "Ratio of vector angles from the nearest to the second nearest neighbor", and gave more informative captions of Fig. 4 as "Relationship between the ratio of vector angles from the nearest to the second nearest neighbor and the number of matching points".

14. Many "of what?" for Fig. 6

The phrases in Fig. 6 have been limit with "of what". Thanks.

15. "Figure 7. Slope piecewise fitting and monotone interval acquisition." of what,please more informative captions.

We added more informative captions as "Figure 7. Slope piecewise fitting and monotone interval acquisition of slopes."

16. "That needs the error correction mechanism to predict/optimize the
216 feature point when it is considered as the tracking point." That ?

We rewrote this sentence as "An error correction mechanism is necessary to predict/optimize the feature point when it is considered as the tracking point."

17. "Figure 8. Images captured during the welding process....."  whats the difference between the images? do we need that much image?

So many images are presented because we want to show the effectiveness of the proposed feature point identification method. These images are captured continuously. There are some differences between them because some little welding slag or spatter is in these images. These images in a red rectangle in Fig. 9 are exactly this case (slag is in these images).

We revised the caption of Fig. 9 as "Feature point identification results in which the images in the red rectangle show the examples of ineffective feature identification because of little welding slag.".

18. For Fig. 9, whats the difference between the images? do we need that much image? How can we see the feature point identification on these images? how can we see the ineffectiveness?

Thanks a lot. We added a sentence to indicate that the feature point is the study object and the tracking position. See Line 221.

Meanwhile, in Fig. 9 we used a blue circle to mark the ineffective identification result and point out the reason why the ineffective result happens. 

19. For Fig. 10, please emphasize wats de difference between fig 9 and these images.

The difference between the two group images is that the feature point identification results in the red rectangle in Fig. 9 is ineffective while the corresponding optimized result in Fig. 10 is effective. That is, the accuracy of the designated tracking position is improved.

20. For Fig. 12, abrevations not defined yet!

To meet this statement, we added the two abbreviations in Lines 102 and 103 in advance. Thanks.

21. For Fig. 14, were is it on the image?

We added "Width" and "Height". However, APWBW  and APWBH represents different width and height of different measurement positions respectively. Therefore, APWBW  and APWBH can not be marked in the image.

22. "We investigate the robustness of the measurement method proposed in this work through changing welding current to obtain the variable WSP during the same welding process." Please rephrase!

We rephrased this sentence as "The robustness of the modeling method proposed in this work is investigated with the variable WSP by changing welding current during the same welding process."

23. For Fig. 18, unify font sizes.

We have checked the font sizes and are sure that the font size is the same. Maybe because of upper and lower case letters, they look different. Thanks.

24. In the caption of Fig. 19......, where are the welding parameters?

As the paper mainly focuses on the vision-based bead geometry feature modeling process, all the related welding process parameters are omitted. We know that the welding current reduce from about 210 A to 170 A in this test, we added it in the brackets. 

25. For Fig. 23, "Sample is misleading, could you write sampling depth or something like that?In preious figs it was sampling result"

We replaced "Sample" with "Sampling time". Thanks!

26. From line 389 to 406. Summary?

Yes. We added the title.

27. conclusions should be somewhat self contained, also please redefine abbrevations here.

We used full names of these abbreviations. 

Thanks again!

Round 2

Reviewer 3 Report

I dont have additional comment.

Reviewer 4 Report

The manuscript was improved significantly from the initial state, the corrections and answers to my questions were adequate, I think the manuscript now satisfy the publication criteria in Sensors.